# Physical Activity Treatment in Adults with Type 2 Diabetes Mellitus per National Treatment Guidelines for Germany: A Telephone-Survey-Based Analysis

**DOI:** 10.3390/healthcare10101857

**Published:** 2022-09-23

**Authors:** Benjamin Wenz, Jonathan Graf, Yong Du, Andrea Teti, Lars Gabrys

**Affiliations:** 1University of Applied Sciences for Sport and Management Potsdam, Am Luftschiffhafen 1, 14471 Potsdam, Germany; 2Institute of Gerontology, Faculty I, Vechta University, Driverstraße 22, 49377 Vechta, Germany; 3Robert Koch Institute, Department of Epidemiology and Health Monitoring, General-Pape-Str. 62-66, 12101 Berlin, Germany

**Keywords:** health care, type 2 diabetes mellitus, telephone survey, physical activity, prevention, public health, epidemiology

## Abstract

Physical activity (PA) is effective in the prevention of type 2 diabetes mellitus (T2DM). According to the German national treatment guidelines for T2DM, PA is recommended at all stages of the treatment process. Adults with T2DM were recruited within the cross-sectional telephone survey ‘Disease knowledge and information needs–Diabetes mellitus (2017)’. Self-reported data on socio-demographic characteristics, previous and current T2DM treatment, and PA behavior were collected. Using multivariable logistic regression models, the correlation between PA treatment (referrals and recommendations) and PA was investigated. Overall, 1149 adults diagnosed with T2DM are included in the analysis. Of the participants, 66.7% reported having ever received PA as part of their T2DM treatment with 61% of the participants reporting PA treatment at the time of the initial T2DM diagnosis and 54% at the time of the interview. Women, older participants, and those with a lower educational level were less likely to have ever been treated with PA. Currently being treated with PA as part of the T2DM treatment was associated with higher rates of achieving the World Health Organization’s PA recommendations (≥150 min per week) (OR = 1.95, 95% CI: 1.42–2.68), as well as ever being treated with PA (OR = 1.74, 95% CI: 1.20–2.38). The analyses showed that PA treatment plays a role in the treatment process of T2DM, but not all patient subgroups benefit in the same way. Efforts to increase PA treatment as part of T2DM treatment are needed, especially for those who are currently not treated with PA. Further research is needed to better understand the type of PA (e.g., structured or unstructured) undertaken by adults with T2DM to develop tailored PA interventions for adults with T2DM and for those in vulnerable subgroups.

## 1. Introduction

Type 2 diabetes mellitus (T2DM) is one of the most frequent chronic metabolic diseases and is characterized by elevated blood glucose levels and acquired insulin resistance or insulin deficiency [1]. T2DM is associated with a significantly increased risk for the development of comorbidities such as cardiovascular diseases, depression, cancer, blindness, kidney failure, or amputation of the lower extremities [2,3,4,5,6]. In addition, T2DM leads to reduced quality of life and life expectancy [7]. T2DM is the most common type of diabetes, accounting for around 90% of all diagnosed DM cases [8]. According to the Health Interview and Examination Survey for Adults (DEGS1), in Germany, 7.5% suffer from any type of diabetes [9]. Based on a population-based survey and claims data, the Robert Koch Institute (RKI), which is the German National Public Health Institute, estimated a lifetime prevalence for T2DM of up to 10% for the adult population in Germany and projected increasing prevalence rates for the next decades [10]. For this reason, the prevention and treatment of noncommunicable diseases, including DM, have become a health priority for Germany in the last decade [11,12]. Many studies have identified several factors associated with the risk of developing T2DM, such as social and contextual factors and lifestyle habits such as smoking and physical inactivity [13,14,15]. Furthermore, research has also shown that T2DM prevention can be achieved through lifestyle intervention [16,17,18]. Different studies have identified physical activity (PA) and a healthy diet as protective factors against the incidence of T2DM [19,20]. International studies have also shown the positive effects of PA on lowering mortality among T2DM patients [21,22,23]. The positive effects of PA and changes in Hb1Ac levels are also known among those with prediabetics [14,15,24].

Based on existing evidence, lifestyle modifications such as PA are usually highlighted as one of the key recommendations in national and international guidelines for the treatment of T2DM [25,26]. In Germany, the National Treatment Guideline (Nationale Versorgungsleitlinie (NVL)) ‘Therapy of Type 2 Diabetes’, recommends lifestyle modifications such as PA and smoking cessation as a cornerstone of T2DM treatment and care [26]. According to the treatment scheme of the NVL, nonpharmacological treatment such as lifestyle interventions (including PA, dietary therapy, and smoking cessation) should be applied at all stages of the treatment process of T2DM. First, patients should be advised to engage in regular unstructured PA to increase their activity level. Second, patients should be motivated and counseled to engage in structured PA and exercise programs in accordance with their individual risk profiles and preferences. Lastly, patients should engage in structured exercise programs with a focus on endurance and/or strength training in addition to pharmacological treatment [26]. To the best of our knowledge, analyses of guideline adherence to PA treatment in T2DM are scarce, and representative data for Germany are missing. In this study, we obtained the descriptive results of a cross-sectional health survey for adults with T2DM in Germany. The objective was to describe the basic characteristics of adults with T2DM with a special focus on (i) sociodemographic factors, (ii) prevalence of PA treatment, and (iii) prevalence of achievement of the WHO’s PA recommendation of ≥150 min per week. Furthermore, we analyzed the association between PA treatment and PA behavior in adults with T2DM.

## 2. Materials and Methods

### 2.1. Study Design

We used the Scientific Use File of the nationwide telephone survey ‘Study on disease knowledge information needs-Diabetes mellitus (2017)’, which was conducted by the Robert Koch Institute (RKI) between September and November 2017 in cooperation with the Office for National Education and Communication on Diabetes Mellitus of the Federal Centre for Health Education (BZgA) and the Institute of Medical Sociology and Rehabilitation Science of the Charité–Universitätsmedizin Berlin. The study design, objectives, and methods as well as results were described in detail elsewhere [27]. In brief, the objective of the telephone survey was to determine the perception of the risks associated with DM, information needs and information-seeking behavior, the personal burden of diabetes disease, and quality of care. All participants were informed about the study’s aims and content and all participants gave informed consent to participate in the study. This paper was written in accordance with the guidelines for reporting observational studies (STROBE statement) [28]. The checklist of items for cross-sectional studies is included in the Appendix A.

### 2.2. Data Collection and Study Population

Data were collected by a standardized interview-based representative telephone survey. Data collection was conducted by the market and social research institute USUMA GmbH (Berlin). Recruiting study subjects comprised two phases. In the first phase, eligibility criteria for participation in the telephone survey were (i) sufficient German language skills, (ii) 18 years of age or older, and (iii) having a landline or a mobile phone number in Germany. This phase yields 2327 adults without diabetes and 263 adults with diagnosed diabetes (response rate: 17.9%). In the second phase, to obtain a large number of adults with DM, a direct screening procedure for adults with diagnosed diabetes was used, which yields 1216 adults with diagnosed diabetes. Thus, the final study sample comprised 1479 adults with self-reported DM aged 18–96 years. Of them, 1333 participants answered the question ‘what type of DM have you been diagnosed with?’ Participants who reported having been diagnosed with type I diabetes mellitus (T1DM) (n = 167) or other types of DM (n = 17) were excluded, resulting in 1149 participants with known T2DM in the present analysis.

For the following analyses, we used self-reported information about the type of diabetes treatment, physical activity, sociodemographic characteristics (age, sex, height, weight, and educational status), and T2DM risk factors.

#### 2.2.1. Diabetes Mellitus Type 2 Treatment

In the survey, five T2DM treatment options were assessed: (i) tablets, (ii) insulin, (iii) other injectable hypoglycemic drugs, (iv) sports or activity, and (v) diet. PA treatment comprised referrals or recommendations from physicians as part of T2DM treatment. Exercise was not directly provided by physicians, and no further information about the type of PA was available. Multiple answers were possible. The question about diabetes treatment was asked at the time of the initial T2DM diagnosis (t0) and in the same way for the current treatment option (t1). Based on this information, we defined four different treatment groups: (i) ever received PA treatment, (ii) PA treatment at t0, (iii) PA treatment at t1, and (iv) never received PA treatment.

#### 2.2.2. Self-Reported Physical Activity and Achievement of WHO PA Recommendation

PA was assessed by using a standardized questionnaire. Participants were asked about their PA level over the past seven days as follows: ‘on how many of the last seven days have you been physically active for at least 30 minutes?’ Achievement of the WHO’s PA recommendations was defined as being physically active for at least 30 min for 5–7 days; being physically active for 0–4 days was categorized as not fulfilling WHO’s PA recommendations [29].

#### 2.2.3. Sociodemographics and Type 2 Diabetes Mellitus Risk Factors

Sex, age, height, weight, and educational status were assessed by self-report. We defined two age groups (≤64 years and ≥65 years) for our analyses. Education was grouped into low, middle, and high based on the parameters defined in the Comparative Analysis of Social Mobility in Industrial Nations (CASMIN) [30]. Furthermore, body mass index (BMI) was calculated by using self-reported weight and height. BMI categorization was defined according to WHO’s definition of normal weight (<25 kg/m^2^), overweight (25–30 kg/m^2^), and obese (≥30 kg/m^2^). Smoking was dichotomously categorized (ever/never). In addition, chronic conditions were assessed by asking participants whether they have ever been diagnosed with (i) cardiac infarction, (ii) stroke, (iii) coronary heart disease, or (iv) depression. Chronic conditions were grouped into two categories: one or more comorbidities and only T2DM with no other comorbidities.

### 2.3. Statistical Analysis

For descriptive and unadjusted analyses, the Rao–Scott chi^2^ test for associations between sex, age, education, BMI, smoking, comorbidities, and achievement of PA treatment was applied. For adjusted analyses, multivariable logistic regression models were applied to account for potential confounding in the PA treatment and PA relationship. The exposure variable was PA treatment (ever vs. never; currently vs. never), with never as the reference category. The outcome of interest was the achievement of the WHO’s PA recommendations (yes/no). Three separate models were run for each outcome. The first model was adjusted only for age and sex, the second model was additionally adjusted for education as a context variable known to be associated with PA behavior, and the third model was additionally adjusted for potentially modifiable risk factors for insufficient PA levels (BMI, smoking, and comorbidities). All analyses were performed using a weighting factor to account for potential deviations between the diabetic sample in the present study and the diabetic population obtained from the ‘German Health Update’ study (GEDA, 2012) in terms of sex, age, and education structure [27]. A *p*-value less than 0.05 was considered statistically significant. All analyses were performed with the statistical software package SAS Studio (SAS Institute, Cary, NC, USA).

## 3. Results

### Sample Description

Table 1 provides the sociodemographic information of the study sample and detailed information on the two groups of participants (ever received PA treatment vs. never received PA treatment). Overall, 66.7% of all participants reported ever having and 33.3% never having received PA treatment. Women were less likely to have ever been treated with PA compared with men (59.7% vs. 74.0%, respectively); BMI, smoking, and the number of comorbidities were negatively associated with PA treatment. Participants with lower education levels reported less frequent PA treatment compared with participants with high educational levels (62.6% vs. 72.9%, respectively). Obese participants reported less frequent PA treatment compared with normal-weight participants (61.8% vs. 73.8%, respectively), and participants with one or more comorbidities were less frequent compared with participants with only T2DM (62.3% vs. 69.9%, respectively).

Overall, 61.0% of all adults with T2DM reported PA treatment at the initial diagnosis (t0) and 54.0% during their current treatment therapy (t1) (Table 2). Little or no differences in PA treatment for the two different time points could be observed for sex, age, educational level, BMI, smoking status, and comorbidities. PA treatment was more common for adults with T2DM during initial diagnosis than during current treatment therapy, independent of the study sample characteristics (Table 2).

Almost half of all adults with T2DM (48.0%) reported PA treatment at the initial diagnosis (t0) and during current treatment (t1). Of those who received PA treatment at initial diagnosis, 12.4% dropped out at current treatment and 6.3% currently received PA treatment but not at initial diagnosis. Of the study sample, 33.3% never received PA treatment at any treatment stage. Table 3 shows that the number of participants who fulfilled the WHO’s PA recommendation was about 20% higher among participants who received PA treatment as part of their current T2DM therapy than among those who did not receive PA treatment (64.0% vs. 45.2%, respectively). In persons who received PA treatment, statistically significant differences in self-reported PA could only be observed for normal-weight persons compared with those who were overweight or obese.

Figure 1 and Figure 2 present the results of the multivariable logistic regression models for self-reported fulfillment of the WHO’s PA recommendation (>150 min per week) adjusted for sociodemographic characteristics (sex, age, educational level, BMI, smoking status, and comorbidities). Being ever treated with PA was associated with a significantly higher chance of the WHO‘s PA recommendation achievement compared with never being treated with PA (OR = 1.74, 95% CI: 1.20–2.38) in the fully adjusted model, as well as currently being treated with PA (OR = 1.95, 95% CI: 1.42–2.68).

## 4. Discussion

So far, there is good evidence for the prevention of T2DM through PA, but in the therapeutic setting, little is known about PA treatment prevalence as part of T2DM treatment because most studies have focused on pharmacological treatment options [31,32,33]. To the best of our knowledge, this was the first analysis showing data on PA treatment in adults with T2DM in Germany. Our findings are based on representative data from a nationwide telephone survey conducted by the RKI in 2017. Even though PA is recommended as a basic therapy in T2DM treatment and one of the key recommendations in the German national guideline for the treatment of T2DM (Nationale Versorgungsleitlinie-NVL) ‘Therapy of Type 2 Diabetes’, our analysis showed that only half of all participants with T2DM report being continuously treated with PA. This is in line with a former report that analyzed data from 2003 in terms of treatment with nonpharmacological lifestyle interventions (e.g., PA treatment or dietary interventions). Therein, the authors concluded that 50% of adults with T2DM had received lifestyle intervention in combination with pharmacological treatment in Germany [34]. However, these data are not fully comparable with our findings because they did not differentiate between dietary and PA treatment, but they did confirm low PA treatment rates in T2DM therapy. This is remarkable because T2DM was the first disease management program (DMP) in Germany and was successfully implemented in 2003. Since 2007, the number of type 2 diabetics subscribed to the DMP T2DM has almost doubled from 2.6 million in 2007 to 4.5 million in 2020 [35]. In 2015, the first German Prevention Act was adopted, which lists the prevention and treatment of T2DM as the highest priority of eight health targets for Germany [36]. Because of these initiatives, PA treatment rates below 50% are not sufficient. Our analyses indicated that not receiving PA treatment was independently associated with sociodemographic characteristics and high levels of social inequality in T2DM treatment. Women with T2DM and vulnerable subgroups (e.g., adults with lower educational status, with higher BMI, smokers, and those with additional comorbidities) were less often treated with PA compared with others. This is in line with previous studies that showed that socioeconomic inequalities in diabetes care still exist [37,38,39]. This is alarming, as persons with T2DM and a higher risk for mortality and cardiovascular disease (e.g., smokers or adults with T2DM with overweight or obesity or with comorbidities) should be given consideration in PA treatment approaches [40].

Furthermore, our analyses showed that PA treatment was associated with significantly higher achievement of the WHO‘s PA recommendation. Our analysis underlined that PA treatment is effective in increasing PA in adults with T2DM, as shown before in a meta-analysis by Figueira et al. [41]. A study by Jordan et al. found that PA counseling by physicians increased participation in PA programs by a factor of 2.5 [42]. Therefore, it seems essential to strengthen national efforts in different settings to promote PA treatment in T2DM therapy and PA counseling, or a combination of both.

Regarding the achievement of the WHO’s PA recommendation, our findings draw an optimistic picture. According to our results, more than 45% of those who did not receive PA treatment and nearly 65% of those who received PA treatment achieved the WHO’s PA recommendations, but this finding should be interpreted with caution. In 2017, Finger et al. found that no more than 45% of the healthy population achieved the WHO’s PA recommendations [43], and a recent study by Sudeck et al. showed that only 34.6% of all adults with diabetes mellitus (type 1 and 2) achieved the WHO’s PA recommendations. Out of all patients with noncommunicable diseases, persons with diabetes were found to be amongst the least physically active [44]. Because Sudeck et al. restricted their analysis to aerobic PA performed for recreation, we used a broader definition of PA in our analysis, which may have led to higher rates of PA. Overall, more research should be conducted to better understand the ideal kind of PA treatment. Moreover, further studies should specify PA preferences of adults with T2DM to develop tailored interventions and treatment measures for this group.

Considering new technologies, such as smartphones and digital devices, strong efforts should be made to explore new and innovative approaches to increasing PA among adults with T2DM. Rapid developments in technology have encouraged the use of health apps in PA research over the past decade. Numerous studies have shown that app-based interventions can be helpful and useful for health promotion and can certainly foster behavioral changes among patients [45,46,47]. Smartphones and fitness trackers can provide therapeutically relevant information on PA behavior and can promote patients’ PA [48,49,50].

Recently, new digital health applications (Digitale Gesundheitsanwendungen, DiGA) have been implemented in health care and can be prescribed by physicians in Germany, with costs reimbursed by health insurance [51]. This might offer opportunities for future health care strategies, but, so far, only two DiGAs for diabetes are listed, and just one is focusing on PA [52].

### Strengths and limitations

A strength of this study was the method of the telephone survey, which ensured a representative sample of all potentially reachable private German households. However, the response rate was low. Furthermore, data ascertainment was performed with standardized and quality-controlled procedures. Because the analysis was restricted to the German-speaking population, the results are not transferable to other populations. All information was based on self-reports, which may have resulted in information bias and social desirability, which both may have led to the overestimation of PA [53]. Moreover, the recommended PA treatment by physicians might vary and, therefore, the impact on the actual PA of adults with T2DM might differ as well. The questionnaire in the survey did not differentiate between the form and type of recommended PA treatment given by physicians and did not collect any information related to possible differences in providing PA treatment. Future studies should therefore request additional information on how physicians provide and what form of PA treatment is given by physicians within the T2DM therapy. It may also be worth examining differences based on survey locations or types of physicians (general practitioners vs. specialists).

In addition, other studies usually distinguished between structured PA programs (e.g., participation in exercise classes) and unstructured PA (e.g., walking, bike riding, gardening, and housework), which we could not in our analyses. This resulted in challenges in assessing the association between PA treatment and PA behavior.

## 5. Conclusions

Overall, our analyses showed that PA treatment is effective as part of T2DM treatment in Germany to enhance PA. Nevertheless, not all patient subgroups benefit in the same way. Efforts to increase PA treatment as part of T2DM treatment are needed for Germany, especially for those who are currently not treated with PA (e.g., women, older, lower educated, and higher BMI). Further research is needed to better understand the type of PA (e.g., structured or unstructured) undertaken by persons with T2DM to develop tailored PA recommendations for adults with T2DM and vulnerable groups. Therefore, healthcare providers should be encouraged to promote PA more often, strengthen patients’ competencies for health, and support patients to increase PA. Although the WHO’s Global Action Plan on PA 2018–2030 states that PA interventions should strengthen peoples’ competencies for health, digital interventions, health apps, and digital devices may offer a potential tool for promoting PA in T2DM treatment [54].

## Figures and Tables

**Figure 1 healthcare-10-01857-f001:**
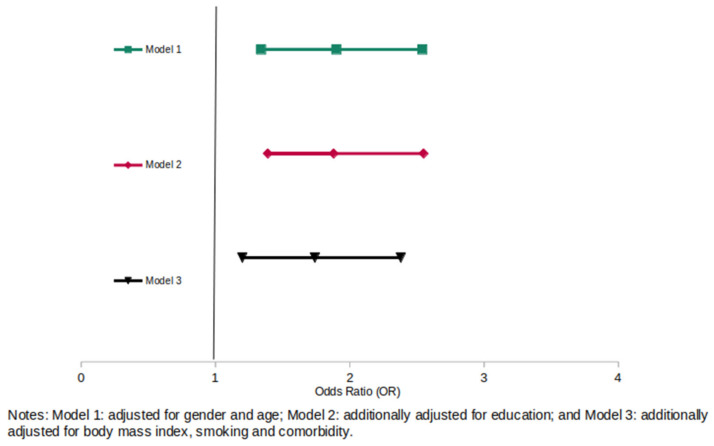
Association of ever receiving PA treatment as part of T2DM therapy and achievement of WHO’s physical activity recommendation to be physically active of at least 150 min/week (n = 1149) compared with never receiving PA treatment.

**Figure 2 healthcare-10-01857-f002:**
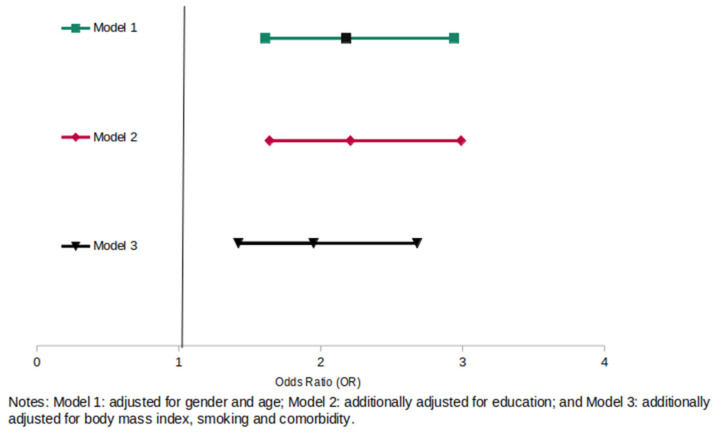
Association of currently receiving PA treatment as part of T2DM therapy and achievement of WHO´s physical activity recommendation to be physically active of at least 150 min./week (n = 1149) compared with never receiving PA treatment.

**Table 1 healthcare-10-01857-t001:** Comparison of persons who ever received physical activity treatment and persons who never received physical activity treatment within the T2DM treatment therapy (n = 1149).

Physical Activity Treatment	Ever (n = 775)			Never (n = 374)			
	%	n	95% CI	%	n	95% CI	*p*-value
Total	66.7	(775)	(63.40–70.05)	33.3	(374)	(29.83–36.75)	
Sex							<0.0001
Male	74.0	(434)	(69.82–78.24)	25.9	(161)	(21.75–30.17)	
Female	59.7	(341)	(54.71–64.67)	40.3	(213)	(35.32–45.28)	
Age group (years)							0.1089
≤64	70.5	(188)	(64.13–76.80)	29.5	(79)	(23.20–35.86)	
≥65	64.2	(562)	(60.3610–68.15)	35.7	(285)	(31.84–39.63)	
Educational level							0.0194
Low	62.6	(206)	(57.07–68.19)	37.4	(132)	(31.80–42.92)	
Average/medium	69.9	(328)	(65.27–74.54)	30.1	(148)	(25.45–34.73)	
High	72.9	(240)	(67.32–78.50)	27.1	(94)	(21.49–32.67)	
Body mass index (kg/m^2^)							0.0144
Normal (<25 kg/m^2^)	73.8	(157)	(66.64–80.93)	26.2	(71)	(19.06–33.35)	
Overweight (25–30 kg/m^2^)	69.7	(321)	(64.50–75.81)	30.3	(127)	(25.18–35.49)	
Obese (≥30 kg/m^2^)	61.8	(297)	(56.65–67.00)	38.2	(176)	(33.01–43.35)	
Smoking							0.0586
Currently smoking	59.4	(98)	(50.50–68.25)	40.6	(64)	(31.74–49.49)	
Not smoking	68.2	(677)	(64.72–71.81)	31.7	(310)	(28.18–35.27)	
Comorbidities							0.0242 *
One or more comorbidities	62.3	(308)	(57.17–67.47)	37.7	(179)	(32.52–42.82)	
No comorbidities only T2DM	69.9	(467)	(65.67–74.25)	30.0	(195)	(25.74–34.32)	

Notes: Given data are weighted percentages (%) and unweighted numbers. * *p* < 0.05. Abbreviation: T2DM = type 2 diabetes mellitus.

**Table 2 healthcare-10-01857-t002:** Characteristics of participants treated with physical activity for 2 different time points (t0 and t1).

	PA Treatment t0 (n = 706)				PA Treatment t1 (n = 648)			
	%	n	95% CI	*p*-value	%	n	95% CI	*p*-value
Total	61.0	(706)	(56.76–64.06)		54.0	(648)	(50.81–58.00)	
Sex				0.0005 *				0.0208 *
Male	67.2	(399)	(62.64–71.85)		58.7	(360)	(53.70–63.84)	
Female	54.9	(307)	(48.92–60.04)		50.2	(288)	(45.06–55.31)	
Age groups (years)				0.0623				0.0551
≤64	65.5	(173)	(58.89–72.18)		59.3	(168)	(52.02–66.55)	
≥65	58.1	(509)	(54.06–62.05)		51.6	(458)	(47.04–55.07)	
Educational level				0.0772				<0.0001 *
Low	57.9	(190)	(52.18–63.67)		45.5	(154)	(40.62–52.48)	
Average/medium	62.5	(291)	(57.64–67.32)		61.4	(285)	(56.56–66.40)	
High	68.1	(224)	(62.29–74.01)		63.4	(208)	(57.31–69.41	
Body mass index (kg/m^2^)				0.1814				0.0151 *
Normal (<25 kg/m^2^)	65.7	(139)	(57.63–73.77)		62.0	(141)	(53.02–70.88)	
Overweight (25–30 kg/m^2^)	63.0	(293)	(57.58–68.40)		58.0	(273)	(52.33–63.65)	
Obese (≥30 kg/m^2^)	57.7	(274)	(52.43–63.00)		48.8	(234)	(43.47–54.16)	
Smoking				0.1131				0.1276
Currently smoking	54.6	(90)	(45.60–63.61)		48.1	(81)	(39.01–57.15)	
Currently not smoking	62.3	(616)	(58.62–66.07)		55.5	(567)	(51.83–59.63)	
Comorbidities				0.0112 *				0.0019 *
One or more comorbidities	55.9	(280)	(50.58–61.15)		47.9	(243)	(42.57–53.13)	
No comorbidities only T2DM	64.7	(426)	(60.27–69.25)		59.2	(405)	(54.40–64.03)	

Notes: Given data are weighted percentage (%) and unweighted numbers. * *p* < 0.05. Abbreviation: T2DM = type 2 diabetes mellitus.

**Table 3 healthcare-10-01857-t003:** Self-reported fulfillment of WHO’s physical activity recommendation (≥150 min per week) of persons who received PA treatment within current treatment therapy (n = 648) and those who did not receive PA treatment (n = 501).

	Received PA Treatment within Current Therapy (n = 648)				Did Not Receive PA Treatment within Current Therapy (n = 501)			
	%	n	95% CI	*p*-value	%	n	95% CI	*p*-value
Total	64.0	(417)	(59.33–68.50)		45.2	(225)	(39.63–50.71)	
Sex				0.5100				0.9003
Male	65.3	(237)	(59.35–71.40)		45.6	(107)	(37.38–53.87)	
Female	62.3	(180)	(55.26–69.29)		44.9	(118)	(37.91–51.96)	
Age groups (years)				0.2983				0.9020
≤64	61.1	(100)	(52.26–69.90)		44.5	(45)	(32.83–56.12)	
≥65	66.5	(306)	(61.24–71.66)		45.3	(173)	(39.44–51.15)	
Educational level				0.2556				0.0858
Low	66.3	(102)	(58.15–74.42)		42.7	(78)	(34.95–5061)	
Average/medium	64.3	(183)	(57.89–70.65)		52.0	(99)	(43.98–60.11)	
High	55.7	(131)	(47.68–63.73)		37.4	(48)	(27.32–47.43)	
Body mass index (kg/m^2^)				0.0152 *				0.6329
Normal (<25 kg/m^2^)	73.5	(102)	(64.22–85.76)		46.1	(44)	(31.55–60.56)	
Overweight (25–30 kg/m^2^)	66.8	(185)	(60.13–73.54)		48.7	(86)	(39.47–57.96)	
Obese (≥30 kg/m^2^)	56.7	(130)	(49.01–64.32)		42.8	(95)	(35.47–50.24)	
Smoking				0.3501				0.9234
Currently smoking	58.4	(52)	(45.22–71.60)		45.8	(36)	(33.08–58.51)	
Not smoking	64.9	(365)	(60.09–6974)		45.11	(189)	(39.23–50.99)	
Comorbidities				0.4485				0.3492
One or more comorbidities	66.2	(159)	(59.14–73.18)		42.6	(99)	(35.11–50.06)	
No comorbidities only T2DM	62.3	(258)	(56.6068.57)		47.7	(126)	(40.03–55.46)	

Notes: Given data are weighted percentages (%) and unweighted numbers. * *p* < 0.05. Fulfilment WHO recommendation: ≥5 days of at least 30 min physical activity. Given data are weighted percentages (%) and unweighted numbers. Abbreviation: T2DM = type 2 diabetes mellitus.

## Data Availability

We used the Scientific Use File of the nationwide telephone survey ‘Disease knowledge and information needs–Diabetes mellitus (2017)’. The datasets used and/or analyzed are available from the RKI upon reasonable request.

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
