# Peer review of "Physical Activity Treatment in Adults with Type 2 Diabetes Mellitus per National Treatment Guidelines for Germany: A Telephone-Survey-Based Analysis"

_healthcare, 2022, doi:10.3390/healthcare10101857_

Round 1
Reviewer 1 Report
The authors analyzed effect of physical activity in T2DM, which is interesting and important.
However, there are some issues in the article:
1. All the results are presented in table. Can the author transform most of the results in figure?
2. There are abundant evidence in the literature focusing on physical activity and T2DM. The author need to address this part carefully in their discussion.
Reviewer 2 Report
This article entitled "Physical activity treatment in adults with type 2 diabetes 2 mellitus in accordance with national treatment guidelines for 3 Germany: A survey-based analysis" is well-organized and easy to read. Good work. However, there are some issues that can be improved for readers' understanding. Please confirm and correct those.
Major concerns:
1. First of all, this reviewer thought that PA treatment was that physicians provided some exercise to patients. But this is not, physicians provided only referrals and recommendations, right? Thus, more detailed information for "not providing exercise" should be needed especially in the Abstract and Method sections.
2. PA treatment seems to be different among physicians providing the referrals and recommendations. For example, one physician could provide PA treatments twice a week with great enthusiasm. But others could not. If it can be different, it is needed to be emphasized in the Method or Limitation sections.
3. Related to comment 2, the difference in providing PA treatment should be considered in your analyses.
Minor concerns:
1. Title: Please state “A telephone survey-based analysis.” As well as in the Abstract section, please emphasize that the telephone was used to collect information.
2. L52: Please spell out PA. This is the first time to show up.
3. Table 1a: The form of tables is different. Please fix it.
4. L186: PA treatment, not PA treat ment.
5. L232: Is “diabetes T2DM” correct?
6. L282: Why do you think “A strength of this study is the large sample size and a representative study sample for Germany”? To this reviewer, the sample size of 1149 is not so large and not enough to say the representative study sample for Germany.
Round 2
Reviewer 1 Report
I have no questions with the current version.
Author Response
Many thanks.
Reviewer 2 Report
Thank you very much for your effort to respond to this reviewer's comments. Clearly, the quality of this paper has been improved. Finally, this reviewer made one query for this article. Please confirm it.
Related to the previous comment 2, this reviewer said "the difference in providing PA treatment should be considered in your analyses."
You responded that "Many thanks for the comment. Unfortunately, the questionnaire does not differentiate between the form of recommended PA treatment. Hence, we can not consider this in our analyses."
Re: I understood your condition for it. However, I think that you can still manage this issue using other items in multivariate analyses, such as survey locations or categorized physician names. If you did not collect the items related to the possible differences in PA treatments, please provide us a solution to the limitation section for future studies not only "However, the questionnaire of the study did not differentiate between the form of recommended PA treatment given by physicians."
Thanks,
Author Response
Many thanks for the comment. We added the argument in the limitations section (line 331-339).